# Multi-Omics Analysis of Genes Encoding Proteins Involved in Alpha-Linolenic Acid Metabolism in Chicken

**DOI:** 10.3390/foods12213988

**Published:** 2023-10-31

**Authors:** Wenjuan Zhao, Yidong Wang, Xiaojing Liu, Yanke Wang, Xiaoya Yuan, Guiping Zhao, Huanxian Cui

**Affiliations:** State Key Laboratory of Animal Nutrition and Feeding, Key Laboratory of Animal (Poultry) Genetics Breeding and Reproduction, Institute of Animal Science, Ministry of Agriculture, Chinese Academy of Agricultural Sciences, Beijing 100193, China; zhaowj0508@163.com (W.Z.); wangyidong1997@163.com (Y.W.); 28810610203@163.com (X.L.); yanke1222@163.com (Y.W.); deletedyxy@163.com (X.Y.); zhaoguiping@caas.cn (G.Z.)

**Keywords:** functional fatty acids (FFAs), regulatory genes, deposition, metabolic pathway

## Abstract

Alpha-linolenic acid (ALA, ω-3) is an antioxidant that reduces triglyceride (TG) levels in blood, a component of cell membranes and a precursor compound of eicosapentaenoic acid (EPA, ω-3) and eicosatrienoic acid (DHA, ω-3). Fatty acid content is a quantitative trait regulated by multiple genes, and the key genes regulating fatty acid metabolism have not been systematically identified. This study aims at investigating the protein-encoding genes regulating ω-3 polyunsaturated fatty acid (PUFA) content in chicken meat. We integrated genomics, transcriptomics and lipidomics data of Jingxing yellow chicken (JXY) to explore the interactions and associations among multiple genes involved in the regulation of fatty acid metabolism. Several key genes and pathways regulating ω-3 fatty acid metabolism in chickens were identified. The upregulation of *GRB10* inhibited the mTOR signaling pathway, thereby improving the content of EPA and DHA. The downregulation of *FGFR3* facilitated the conversion of ALA to EPA. Additionally, we analyzed the effects of ALA supplementation dose on glycerol esters (GLs), phospholipid (PL) and fatty acyl (FA) contents, as well as the regulatory mechanisms of nutritional responses in FFA metabolism. This study provides a basis for identifying genes and pathways that regulate the content of FFAs, and offers a reference for nutritional regulation systems in production.

## 1. Introduction

Due to its lack of certain enzymes, such as esterases and fatty acid dehydrogenases, the body is unable to directly synthesize some functional fatty acids (FFAs) on its own. In general, alpha-linolenic acid (ALA, omega-3 (ω-3)) can be produced as eicosapentaenoic acid (EPA, ω-3) under the joint action of fatty acid elongase and fatty acid dehydrogenase, followed by docosahexaenoic acid (DHA, ω-3), after dehydrogenation and extension [1]. ALA can also be synthesized by acetyl coenzyme A desaturase to produce eicosapentaenoic acid (EPA) from docosahexaenoic acid (DHA), which are then oxidized through β-oxidation [2,3]. Therefore, in animals, ω-3 fatty acids, such as ALA, and omega-6 (ω-6) fatty acids, such as linoleic acid (LA), cannot be synthesized by themselves and need to be taken in through feeding. In addition, they are involved in many vital functions of the body; thus, they are called FFAs. ALA and its metabolites EPA and DHA have been found to play an important role in cell membrane permeability, fluidity and the transport of substances [4]. As cellular active molecules, they can also play an important role in the body’s health [5]. Studies have shown that the efficiency of ALA’s conversion to EPA and DHA is limited, and there is some competition with LA; thus, the ω-6/ω-3 ratio is reduced by dietary supplementation with ALA to protect the health of animals [1]. However, the nutritional regulation mechanisms of ω-3 polyunsaturated fatty acids (PUFAs) remain unclear.

A study conducted identified several candidate genes associated with fatty acid deposition in chicken breast meat using a genome-wide association study (GWAS) [6]. They identified a region on chromosome 6 that is strongly associated with the deposition of several fatty acids, including palmitic acid, oleic acid and LA. In another study, Liu et al. performed a transcriptome analysis to identify differentially expressed genes (DEGs) associated with fatty acid deposition in chicken liver [7]. They identified several DEGs involved in lipid metabolism, including *ACACA*, *FASN* and *SCD*, which play a key role in fatty acid synthesis. The potential regulation of fatty acid metabolism in the treatment of central nervous system diseases has been studied via lipidomics analysis, revealing that *SGLT2* reduces blood glucose levels, as well as the symptoms of muscle loss [8,9]. OPA1 mitochondrial dynamin-like GTPase (OPA1) is a cell membrane-bound kinetic protein GTPase involved in lipid remodeling that is used in lipidomics to reveal impaired fatty acid fluxes, and thus, triglyceride (TG) accumulation in fibroblasts [10]. In recent years, joint multi-omics analyses have enabled the screening of genetic loci for important metabolic traits to understand the genetic basis of metabolic diversity and its correlation with complex traits. Many studies have shown that combined multi-omics analyses can reveal the effects of different types and levels of fatty acids in poultry feed on metabolism and the regulatory mechanisms of different metabolic pathways and differential gene expression [11]. Through combined multi-omics analysis, the expression patterns of genes encoding fatty acid-metabolizing enzymes (such as *FAS* and *ACOX*) and transcription factors (such as *LXR* and *PPAR*) were found to correlate with the content and composition of PUFAs in poultry [12]. In addition, combined multi-omics analysis can also be used to identify key genes and biomarkers in poultry breeding, thus providing important scientific support for poultry breeding and nutrition regulation.

In China, agriculture is the foundation of the country, and the foundation of the country is solid [13]. With the introduction of the big food concept in 2015, to make China’s rice bowl better and better, improving the nation’s healthy eating habits has become a pressing problem that needs to be solved. Currently, China is a leading country in meat production and consumption, and chicken production has hugely contributed to the increase in meat products [14,15]. Therefore, improving the quality of chicken meat has become a prevalent problem that researchers and producers need to address together. Therefore, the study of ALA and its metabolism is the focus of current research [7]. In this study, a GWAS was performed based on fatty acid content and genomic data to preliminarily identify the candidate genes and pathways that regulate the content of ω-3 PUFAs. This was followed by supplementation with exogenous ALA and the use of lipidomics analysis to determine changes in lipid deposition and fate. Transcriptome key genes were identified via DEG analysis and weighted gene co-expression network analysis (WGCNA). Finally, the key genes regulating the content of ω-3 PUFAs were identified using a combined multi-omics analysis to provide a theoretical basis for investigating the genes and pathways that regulate the content of ω-3 FFAs such as ALA.

## 2. Materials and Methods

### 2.1. Animals and Sample Collection

The experimental population for this study consisted of 477 chickens of a slow-growing selection line of Jingxing yellow (JXY) broiler chickens provided by the Chinese Academy of Agricultural Sciences (Beijing, China). These chickens were kept at an ambient temperature of 15–27 °C, relative humidity of no less than 50% and an average light duration of 16 h, until slaughtered at 98 days of age. Detailed information on animal management is described elsewhere [6,7]. Pectoral muscle tissue and blood were collected for the genome-wide association study (GWAS).

### 2.2. Design of Animal Diets

A total of 300 chickens (half males and half females) were randomly selected for treatment involving supplementation with ALA. The chickens were randomly divided into three groups: basal diet feeding (NC), basal diet + 1% ALA supplementation feeding (ADD1) and basal diet + 2% ALA supplementation feeding (ADD2), and the nutrient composition is shown in Table 1 [16]. Five replicates were performed for each group, with 20 chickens (10 males and 10 females) in each replicate. After feeding from one day of age until 56 days of age, 30 individuals of uniform weight were selected from each group and slaughtered. Samples were treated in the same way as JXY chickens.

### 2.3. Genome Sequencing and Genome-Wide Association Study

Detailed information on genomic DNA extraction and sequencing is available elsewhere [14]. A DNA library was constructed for each sample, and at least 10 G of raw sequencing data (raw data) was obtained for each sample. Filtered clean reads were aligned to the *Gallus gallus* (chicken) reference genome (GRCg6a: GCA_000002315.5) using the MEM mode of BWA software (version 0.7.12) (http://bio-bwa.sourceforge.net, accessed on 10 July 2021)). Picard tools (version 1.119) (https://broadinstitute.github.io/picard/, accessed on 10 July 2021) and SAMtools (version 1.9) (http://samtools.sourceforge.net/, accessed on 10 July 2021) were then used to generate sequenced BAM files. The Genome Analysis Toolkit (GATK, version 4.0.2.1) was used to perform variant calling and variant filtering, and their quality control and filtering conditions were reported in a previous study [17]. Clean DNA sequencing data were deposited in the Genome Sequence Archive (GSA) at the BIG Data Center at Beijing Institute of Genomics (BIG) of the Chinese Academy of Sciences (https://bigd.big.ac.cn/gsa/, accessed on 10 July 2021) (Member, 2019) under accession numbers CRA002643 and CRA002650 [17,18,19]. Data are publicly accessible at http://bigd.big.ac.cn/gsa, accessed on 10 July 2021.

The single-trait mixed linear model (MLM) of Genome-Wide Efficient Mixed-Model Analysis (GEMMA) software was used to perform the GWAS for the expression levels of key genes. The statistical model was as follows:y=α+ Xβ+μ+ ε
where y is a vector of phenotypic values, α is the corresponding coefficient including the intercept, X is a vector of genotypes, β is a vector of single-nucleotide polymorphism (SNP) effects, μ is a random effect and ε is the residual.

The threshold for reaching a significant 5% *p*-value at the genomic level after Bonferroni correction was 7.58 − log10 (0.05/1,891,471), and the threshold for reaching a potentially relevant *p*-value at the genomic level was 6.23 − log10 (1/1,891,471). The GWAS results were visualized as Manhattan and quantile-quantile (q-q) plots generated using the qqman package in R software (version 4.1.0). Then, ChipPeakAnno was used to annotate these loci lines.

### 2.4. Determination of Fatty Acid Content and Composition in Chicken Meat

Fatty acid content and composition in chicken meat was determined via gas chromatography (GC). Specific methods for determining fatty acid content and composition were reported previously [6].

### 2.5. Determination of TG and PL Contents in Chicken Meat

Pectoral muscle tissue homogenates were obtained by stripping the entire right pectoral muscle, removing the fat and fascia and churning the sample on ice using a meat grinder. The TG content was determined using a TG kit obtained from the Nanjing Jiancheng Institute of Biological Engineering (Nanjing, China), according to the manufacturer’s instructions. The PL content was determined using a PL assay kit obtained from Beijing Lidman Biochemical Co., Ltd. (Beijing, China).

### 2.6. Chicken Lipidome Extraction and Analysis

Lipids from 40 samples were extracted using the methyl tert-butyl ether (MTBE) method [20]. The results of the analysis were obtained at Shanghai Applied Protein Technology Co., Ltd. (Shanghai, China). Lipid species were identified using Thermo Scientific™ LipidSearch™ software version 4.2 (Thermo Fisher Scientific Inc., Waltham, MA, USA) to process the raw data. For the data extracted using LipidSearch, ion peaks with a value of >50% missing from the group were removed. After normalization and integration using the Perato scaling method, the processed data were imported into SIMCA^®^-P 16.1 (Sartorius Stedim Data Analytics AB/Umetrics, Umea, Sweden) for multivariate statistical analysis, including principal component analysis (PCA), partial least squares discriminant analysis (PLS-DA) and orthogonal partial least squares discriminant analysis (OPLS-DA). Lipids with significant differences were identified based on a combination of statistically significant thresholds of variable influence on the projection (VIP > 1) values obtained from the OPLS-DA model (muti-dimensional statistical analysis) and a two-tailed Student’s *t*-test (*p* value < 0.05) on the raw data (unidimensional statistical analysis) using PASS 16 (https://www.ncss.com/software/pass/, accessed on 23 December 2022) before the experiments.

### 2.7. Pectoral Muscle RNA Extraction and Sequencing

Total RNA samples were extracted from 10 hens in the ADD1 group and 10 hens in the NC group, and sequenced by Shanghai Applied Protein Technology Co. (Shanghai, China). The data generated using the Illumina platform (Illumina Inc., San Diego, CA, USA) were first evaluated using FastQC software, and the read junctions were removed using BBMap software (version 38.16). Then, the clean data were obtained by filtering the unqualified reads using Fastp. The sam files were converted to bam files using SAMtools software (v 1.12), and then, sorted, input to QC and indexed. Finally, genes were quantified using the FeatureCount software, and gene expression was normalized using the DESeq2 package in R (v 4.1.1).

### 2.8. Identification of DEGs

Ten individuals each with high or low differences in FFA content were selected, and differentially expressed transcripts were analyzed using the DESeq2 package in R (v 4.1.1). Genes with *p* < 0.05 and |log2foldchange| > 1 were considered to be DEGs.

### 2.9. WGCNA of Candidate Genes

The samples were clustered using the hclust function of the R (V 4.1.1) WGCNA package to eliminate outliers. The best soft threshold was then selected using the pick-SoftThreshold (sft) function. A topological overlap matrix (TOM) was then constructed using the adjacency function, and the genes were clustered using the hclust function according to the similarity of the TOM matrix. The eigenvalues (Module eigengene, ME) of each module were calculated using PCA, and the modules with similarity greater than 0.7 were merged according to the ME for clustering. The ME of the merged modules was recalculated and correlation analysis was performed with the phenotype matrix to select the module with high correlation with the phenotype as the target module. Subsequently, module membership (MM) and gene significance (GS) within the target modules were calculated, and the genes meeting the screening criteria (|MM| > 0.2 and |GS| > 0.8) were considered as hub genes.

### 2.10. Functional Annotation of Candidate Genes

The Kyoto Encyclopedia of Genes and Genomes (KEGG) databases were used in the KEGG Orthology-Based Annotation System (KOBAS) (http://kobas.cbi.pku.edu.cn/, accessed on 13 February 2023). Candidate genes were subjected to KEGG functional annotation and functional set enrichment analysis (http://kobas.cbi.pku.edu.cn/, accessed on13 February 2023). Genes and Genomes (KEGG), a large and widely used database for gene enrichment analysis, was used for the functional enrichment of candidate genes. Significantly enriched pathways were also screened at *p* < 0.05, and the entries of significantly enriched pathways were visualized using the clusterProfiler package in R.

### 2.11. Data Statistics and Analysis

Batch correction was performed on the fatty acid phenotype data. R studio in R software version 3.6.1 was used to remove the batch effects on phenotypes using a generalized linear model (GLM). Phenotypic data were analyzed using Excel 2016 (Microsoft Corporation, Redmond, WA, USA). Data are expressed as “mean ± standard deviation”. Correlation analysis was performed using R software (version 3.6.1). One-way analysis of variance (ANOVA) was performed using GraphPad Prism 8.3 (GraphPad Software Inc., San Diego, CA, USA), and a significant difference between the two groups was considered at *p* < 0.05.

## 3. Results

### 3.1. Genomic Screening of Candidate Genes Possibly Involved in Functional Fatty Acid Metabolism

We generated sequencing data for 477 JXY chickens with an average coverage depth of 10×. These data were used to identify genome-wide-level variants, and a total of 8,777,521 SNP datasets were generated.

A GWAS was then performed based on MLM association analysis in 477 JXY chickens using three FFAs, namely ALA, EPA and DHA (Figure 1). Unfortunately, only five potential SNP loci were detected in ALA (Table 2). In order to obtain more candidate genes, the GWAS results were sorted by *p* value from lowest to highest to identify the top 0.1% loci. The results showed that the variant loci associated with ALA content in chicken meat were annotated with 760 genes, including *HADHB*, *ELOVL5*, *HADHA*, etc. The variant loci associated with EPA content in chicken meat were annotated to 903 genes, including *ACSL1*, *ACSL4*, *ELOVL5*, etc. The variant loci associated with DHA content in chicken meat were annotated to 717 genes (Schedule S1), including *DGKI*, *AKR1E2*, *MBOAT1*, etc.

KEGG enrichment analysis was performed on genes identified and annotated by GWAS. A value of *p* < 0.05 was used as the enrichment condition to identify significantly associated pathways (Schedule S2). Candidate genes associated with ALA content in chicken were enriched in 128 pathways, including some of interest, such as Fatty acid elongation, the regulation of lipolysis in adipocytes, fatty acid degradation, the MAPK signaling pathway and phosphatidylinositol signaling system pathways. Candidate genes associated with EPA content in chicken meat were enriched in 123 pathways, including some of interest, such as fatty acid biosynthesis, the PPAR signaling pathway, the MAPK signaling pathway, the mTOR signaling pathway and the phosphatidylinositol signaling system. Candidate genes associated with DHA content in chicken meat were enriched in 117 pathways, including some of interest, such as the MAPK signaling pathway, the mTOR signaling pathway, the phosphatidylinositol signaling system, etc.

The finding that the MAPK signaling pathway and phosphatidylinositol signaling system were enriched in all three fatty acid candidate genes suggests that they may be important pathways involved in regulating FFA metabolism. On the other hand, the mTOR signaling pathway was only enriched in the candidate genes regulating EPA and DHA content. The mTOR signaling pathway was found to be associated only with the candidate genes regulating EPA and DHA contents, and may be an important pathway involved in regulating EPA and DHA contents.

### 3.2. Effect of Supplementation with ALA on Production Performance, and Content of PL, TG and Fatty Acid

In order to further establish the function of the candidate genes identified by the previous GWAS screening, the method was validated through exogenous dietary supplementation of different concentrations of ALA. During the experiment, all individuals tested were in good health, with no morbidity or mortality and good feeding management. The results of the spherical test showed no interaction between sex and the supplementation with different concentrations of ALA, so the sex factor was ignored. The supplementation of different concentrations of ALA had no significant effect on body weight and TG content (*p* > 0.05) (Figure 2, Table 3). The effect of supplementation with different concentrations of ALA on PL in chicken meat is shown in Table 3. As the concentration of ALA increased, the PL content showed a trend of increasing first, and then, decreasing. The PL content of the ADD1 group was significantly higher than that of the ADD2 group (*p* < 0.05) and that of the NC group (*p* < 0.01), but there was no significant difference between the PL content of the ADD2 and NC groups.

The results also show that there was no reciprocal effect of sex on the fatty acid composition of chicken meat (*p* > 0.05), indicating that there was no significant effect of sex on the fatty acid composition of chicken meat (*p* > 0.05). The results of fatty acid content in chicken meat after the addition of different concentrations of ALA are shown in Table 4. The ratio of ω-6/ω-3 PUFAs decreased significantly (*p* < 0.001) from 12.21 to 2.42 with an increase in ALA addition. The levels of AA in chicken meat in the ADD1 and ADD2 groups were significantly lower than those in NC group and decreased gradually with an increase in the concentration of ALA added from 2.29 to 1.42 mg/g. The levels of ALA, EPA and DHA in the ω-3 PUFAs all increased significantly with an increase in the concentration of ALA added (*p* < 0.001). In particular, the level of ALA in chicken meat increased from 0.318 to 2.571 mg/g; in the ADD1 and ADD2 groups, it increased 4.39-fold and 8.09-fold, respectively, compared to the NC group, while in the ADD2 group, it increased 1.54-fold compared to the ADD1 group. The level of EPA in chicken meat increased from 0.15 to 0.58 mg/g; in the ADD1 and ADD2 groups, it increased 2.98-fold and 3.89-fold, respectively, compared to the NC group, while in the ADD2 group, it increased 1.88-fold compared to the ADD1 group. The level of DHA increased from 0.30 to 0.68 mg/g in the control group; in the ADD1 and ADD2, it increased 2.07-fold and 2.23-fold, respectively, compared to the NC group, while in the ADD2 group, it increased 1.12-fold compared to the ADD1 group. These findings indicate that an increase in ALA concentration in chicken meat after dietary supplementation with ALA led to an increase in ω-3 PUFA production with ALA as a substrate and a decrease in ω-6 PUFA binding with LA as a substrate, which, in turn, resulted in an increase in ω-3 PUFA levels and a decrease in ω-6 PUFA levels.

The changes in PL content and fatty acid composition after dietary supplementation with ALA showed that the dietary supplementation with ALA was more effective in the ADD1 group; thus, subsequent lipidome and transcriptome analyses were performed in the ADD1 and NC groups.

### 3.3. Effect of Dietary Supplementation with ALA on Lipid Composition

The peaks of the lipid molecules were identified and extracted using LipidSearch software (v 16.1), which identified a total of 3576 lipid metabolites. These 3576 lipid metabolites were classified into 8 major classes and 45 subclasses. The data extracted by LipidSearch were then evaluated for quality, and the data that passed quality control were analyzed. The data were normalized according to the median, Log10 transformation and partial least squares using the MetaboAnalyst (V5.0) platform for anions and cations, independently. A comparison of the lipid profiles of chicken breast samples using unsupervised PCA and OPLS-DA revealed that the ADD and NC groups showed clear separation in the anion and cation modes, as shown in Figure 3A,B. The plot of OPL-SDA scores also showed clear separation between the ADD1 and NC groups (Figure 4), with values of 0.981 and 0.920 for R^2^ and Q^2^, respectively. These results indicate the stability of the developed model and its excellent ability to explain and predict the differences between samples. Additionally, the interaction test performed revealed that ADD1 and NC were not affected by sex, as shown in Figure 3C, D. Therefore, the effect of the sex factor was not considered in the subsequent analysis.

After determining that there were differences in lipid profiles between the two groups, and calculating the VIP values based on the constructed PLS-DA model, the conditions of |Log2Fold Change| > 1, VIP > 1 and *p* < 0.01 were selected to screen for differential lipid metabolites and identify potential markers among them. A total of 206 and 272 differential lipid metabolites were screened in the anionic mode and cationic mode, respectively, which were then classified according to their structures (Figure 5). A total of 20 differential lipid metabolites were identified in the cationic mode, among which phosphatidylcholine (PC), phosphatidylethanolamine (PE), TG, diglyceride (DG), and cardiolipin (CL) were the top five lipids. A total of 19 differential lipid metabolites were identified in the anionic mode, among which PC, PE, CL, phosphatidylserine (PS) and phosphatidylinositol (PI) were the top five lipids. These results suggest that there are significant differences in lipid profiles between the ADD1 and NC groups.

In particular, in the anionic and cationic modes, there were 453 differentially upregulated lipid metabolites and 25 downregulated lipid metabolites after dietary supplementation with ALA compared with the NC group (Figure 6), which shows that most of the lipid components were upregulated after dietary supplementation with ALA. Therefore, it can be inferred that dietary supplementation with ALA led to lipid deposition.

Additionally, the screening for differential lipids containing C18:3 identified a total of 62 differential lipids containing C18:3, including 27 TGs, 6 PSs, 2 PGs, 9 PEs, 13 PCs, 3 CLs, DG, PI and Acetyl-CoA carboxylase (AcCa). The screening for differential lipids containing C20:5 identified 58 differential lipid substances, including 21 TGs, 15 PCs, 7 PEs, 4 PI, 3 PSs, 2 lysophosphatidylcholines (LPCs), hexosyl-ceramide (HexCer), FA, DG, PG, lysophosphatidylethanolamine (LPE) and AcCa. The screening for differential lipids containing C22:6 identified a total of 53 differential lipids, including 9 TGs, 16 PEs, 16 PCs, 5 PSs, 2 PIs, 2 DGs, LPE, LPC and FA. After the removal of concurrently present differential lipid metabolites, a total of 150 differential lipids were identified, including 43 TGs, 40 PCs, 28 PEs, 11 PSs, 7 PIs, 3 PGs, 3 CLs, 3 LPCs, 4 DGs, 2 AcCas, 2 FAs, 2 LPEs and HexCer. It is noteworthy that all of the above differential lipids were upregulated. Taken together, these results indicate that dietary supplemented ALA and its metabolites EPA and DHA were incorporated into glycerophospholipids (GPs), glycerol esters (GLs) and fatty acyl (FA), respectively.

### 3.4. Screening for Differentially Expressed Genes Encoding Proteins Affecting FFA Metabolism

The processed transcriptome sequencing data were aligned and mapped to the chicken (*Gallus gallus*) reference genome (version GRCg6a), which ultimately detected 22,096 genes. Since the fatty acid composition study results show significant differences in ALA, EPA and DHA contents between the ADD1 and NC groups, differential gene expression analysis was performed directly between the ADD1 and NC groups. Using |Log_2_Fold Change| > 1 and *p* < 0.05, as the screening criteria, a total of 1563 DEGs were identified using DESeq2 package analysis, among which there were a total of 609 upregulated genes and 954 downregulated genes, which are shown in Figure 7A and the volcano plot in Figure 7B.

These DEGs were also analyzed via KEGG pathway enrichment analysis, and each pathway was ranked from the lowest to highest *p* value, and the top 20 pathways were plotted as shown in Figure 7C. The DEGs were mainly enriched in KEGG pathways involved in lipid metabolism, including fatty acid metabolism (*EHHADH*, *ACSL6*, *ACSBG1*, *ACSL4*, *ELOVL1*), fatty acid degradation (*EHHADH*, *ACSL6*, *ACSBG1*, *ACSL4*), fatty acid biosynthesis (*ACSL6*, *ACSBG1*, *ACSL4*), the adipocytokine signaling pathway (*IRS2*, *ACSL6*, *ACSL4*, *NFKBIE*, *ACSBG1*, *PPARA*), the PPAR signaling pathway (*PLIN2*, *ACSL6*, *ACSL4*, *ACSBG1*, *PPARA*, *EHHADH*, *MMP1*), the MAPK signaling pathway (*TGFB2*, *RASGRP1*, *HSPA8*, *EPHA2*, *MAP2K3*, *AKT3*, *MAPK11*, *PLA2G4B*, *IL1B*, *PGF*, *MECOM*, *GADD45B*, *CACNA1C*, *MAP3K4*, *FGFR3*, *MYC*, *CACNA1H*, *FGF6*, *DUSP10*, *PTPN5*, *FGF2*, *RPS6KA3*, *MET*) and 20 additional pathways.

### 3.5. WGCNA Screening for Candidate Genes Regulating FFA Content

The analysis of the transcriptome data was restricted to genes expressed in the top 75% absolute deviation of the median, retaining 16,419 genes for subsequent analysis. Clustering analysis was performed on 20 samples. The optimal soft threshold sft = 4 for R^2^ > 0.85 was determined, as shown in Figure 8. The clustering was performed according to the gene expression of different samples, and the modules with correlations > 0.75 were merged to ultimately obtain 14 correlation modules, as shown in Figure 9. Genes with insignificant co-expression trends were uniformly grouped into Grey modules. Correlation analysis was performed on the phenotype matrix, and the correlation coefficients and *p* values of each module with FFA content were obtained, as shown in Figure 10.

Four of the modules (grey 60, light green, brown and purple) were positively correlated with the ALA, EPA and DHA contents (r > 0.45, *p* < 0.05), whereas the light cyan module was significantly negatively correlated with the ALA and DHA contents (r < −0.49, *p* < 0.05), and the tan module was only significantly negatively correlated with the ALA content (r = 0.47, *p* < 0.05) (Figure 10).

The key genes of the modules were identified according to the criteria of |GS| > 0.2 and |MM| > 0.8. The identified key genes in five FFAs included 384 key genes related to ALA content, 1647 key genes related to EPA content, and 1412 key genes related to DHA content.

KEGG pathway enrichment analysis was performed for the key genes of these three traits, and it identified significantly associated pathways (Schedule S3). Nine pathways were jointly enriched for these three traits, namely, autophagy-animal, nucleocytoplasmic transport, oxidative phosphorylation, the mRNA monitoring pathway, protein processing in the endoplasmic reticulum, the spliceosome, the mTOR signaling pathway, oocyte meiosis, endocytosis, and the insulin signaling pathway.

### 3.6. Combined Multi-Omics Analysis to Identify Candidate Genes Regulating FFA Content

The genes mapped to Top 0.1% in JXY chickens, DEGs and hub genes were taken as the intersection, and the genes that were present at the same time were defined as candidate genes. There were 7, 14 and 10 candidate genes related to ALA, EPA and DHA contents, mapped in JXY chickens, and the results are summarized in Table 5.

KEGG enrichment analysis of the candidate genes was performed to identify the top 10 pathways. The pathways found to be associated with ALA content in JXY chickens were the spliceosome (*DDX5*), actin cytoskeleton regulation (*FGFR3*), endocytosis (*FGFR3*), and the MAPK signaling pathway (*FGFR3*). The pathways found to be associated with EPA content in JXY chickens were the mTOR signaling transduction pathway (*GRB10*, *PIK3CB*, *ULK1*), autophagy-animal (*PIK3CB*, *ULK1*), regulation of the actin cytoskeleton (*PIK3CB*, *FGFR3*), endocytosis (*GRK3*, *FGFR3*) nitrogen metabolism (*CA4*), the hedgehog signaling pathway (*GRK3*), the VEGF signaling pathway (*PIK3CB*), mitochondrial-animal (*ULK1*), the metabolism of phosphatidylinositol (*PIK3CB*), progesterone-mediated oocyte maturation (*PIK3CB*) and 10 additional pathways. The pathways associated with DHA content in JXY chickens were the cell cycle (*RB1*), adrenergic signaling in the heart (*CREM*), the mTOR signaling pathway (*GRB10*), cellular senescence (*RB1*) and four additional pathways.

Functional enrichment analysis revealed that the MAPK signaling pathway was enriched in the above-mentioned genomic candidate genes. Thus, this pathway was also screened for the candidate genes that regulate ALA content in chicken meat during the multi-omics joint analysis of candidate genes. The results suggested that the MAPK signaling pathway may be the key pathway regulating ALA, and *FGFR3* may be the key gene regulating ALA and EPA contents. In addition, the mTOR signaling pathway was found to be involved in both EPA and DHA content in JXY chickens. Thus, the MAPK signaling pathway may be a key pathway regulating EPA and DHA. The gene GRB10 was found to be among both candidate genes regulating EPA content and candidate genes regulating DHA content in chicken, and may be a key gene regulating both EPA and DHA content.

## 4. Discussion

Most of the PUFAs in chicken are obtained from dietary sources. The consumption of PUFAs can improve fatty acid composition and be a source of supplementation of the essential fatty acids needed by the body [21]. The main ω-3 fatty acid in pectoral muscle is EPA produced through the elongation of ALA. Therefore, dietary supplementation with ALA will have a beneficial effect on the organism. In this study, supplementation with different concentrations of ALA was performed in compliance with the standard feeding management model. It was found that with a increase in the ALA content, the fatty acid composition profile was changed without altering the production performance, which resulted in a significant increase in fatty acid content in chicken meat, a result that is consistent with the trend observed in previous studies, as well as the reported changes in fatty acid content and composition in serum [22,23,24]. Perilla seeds are rich in ALA, and supplementing feed with perilla seed oil was found to increase the ω-3 content of chicken meat and decrease the ω-6/ω-3 ratio, while decreasing the antioxidant level of the animal [25]. The addition of fatty acids to feed can also contribute to meeting the energy needs of poultry, improving performance in production, and reducing the level of abdominal fat [26,27]. According to dietary recommendations for humans, broiler meat containing high levels of these fatty acids can be a good alternative source of these fatty acids in the human diet [28].

The ω-6 and ω-3 fatty acids are competitively metabolized by the same group of desaturases, elongases and oxidases. Therefore, the lipids produced via common oxidative metabolism pathways tend to be antagonistic [29]. This is consistent with the trend of one increase and one decrease in ALA and LA with increasing substrate concentration (increasing concentration of added ALA) in this study, and AA, EPA and DHA also conform to this pattern. Increasing the intake of LA can reduce the concentrations of EPA and DHA, indicating that dietary LA inhibits the biosynthesis of ω-3 fatty acids, while dietary ALA can affect the biotransformation of ω-6 in the liver [30].

The ω-6/ω-3 ratio can have certain effects on both humans and animals [31]. For instance, a high ω-6/ω-3 ratio can lead to cardiovascular disease and affect lipid metabolism, and a low ω-6/ω-3 ratio can also affect the expression of related genes and reduce the invasive ability of cancer cells [32,33]. In chicken, maintaining a low ω-6/ω-3 ratio also promotes an increase in the EPA content of chicken meat, which is better utilized by consumers [34]. In this study, the ω-6/ω-3 ratio was 12:1 without the addition of ALA, and the addition of 1% ALA and 2% ALA resulted in a decrease in the ω-6/ω-3 ratio with increasing concentrations of ALA. Several studies have shown that ω-6/ω-3 ratios close to 4:1 have more beneficial effects on the organism and make it healthier [31,35,36], and the ratio of ω-6/ω-3 in the ADD1 group in this study was closer to 4:1, so the addition of a 1% concentration of ALA would more likely have a beneficial effect. The results of this study support the theory of nutritional regulation.

Since the ω-6/ω-3 ratio in the ADD1 group was closer to 4:1, the lipid group was also selected for lipid group determination and subsequent histological analysis of chickens in the ADD1 group supplemented with 1% ALA. The results showed that fatty acids obtained from the diet can be part of cell membranes both as PLs and reserve lipids as TGs [37]. Defries et al. found selective incorporation of ALA into PLs, supporting the metabolic hierarchy of ALA incorporation into specific PLs [38]. The intake of an ALA-rich diet can affect the composition of PL fatty acids in the serum of patients with metabolic syndrome [39]. More importantly, PLs and TGs are components of intramuscular fat, and PUFAs are often available after the degradation of intramuscular fat [40,41]. Kong et al. found that an increase in PL molecules may reduce the production of woody and white-striped meat, suggesting that an increase in PLs also has a positive effect on meat quality [42]. In this study, ALA, EPA and DHA were found to be mainly present in GP and GL, followed by their breakdown to free fatty acids. The findings of the present study are consistent with those of previous studies and explain that dietary supplementation with ALA leads to an increase in the content of ALA, EPA and DHA and the deposition of lipids to GP and GL.

The GRB10-encoded protein (growth factor receptor bound protein 10) is a direct substrate in the mTOR signaling pathway and directly inhibits mTORC1 through a phosphorylation-dependent feedback mechanism [43]. The mTOR signaling pathway is a key signaling pathway that regulates cell proliferation and metabolism, and also plays an important role in fatty acid metabolism. During fatty acid synthesis and catabolism, the mTORC1 pathway can affect fatty acid metabolism by regulating the expression of fatty acid synthase and fatty acid oxidation-related genes [44]. At the same time, the mTORC2 pathway can also be involved in the regulation of fatty acid synthesis and oxidation processes, as well as the growth and differentiation of muscle and adipose tissue [45]. In addition, the mTOR signaling pathway can influence the activity of immune cells related to lipid metabolism, and the development of metabolic diseases, such as diabetes and obesity [45]. Therefore, we hypothesized that *GRB10* upregulation and inhibition of the mTOR signaling pathway would result in elevated levels of EPA and DHA content.

In this study, genes encoding proteins affecting ALA and EPA content were enriched in the mTOR signaling pathway (*GRB10*) and MAPK signaling pathway (*FGFR3*). *FGFR3* promotes fatty acid synthesis and desaturation in a PI1K-mTORC1-dependent manner through cleavage and the activation of *SREBP1*, which promotes adipogenesis [46]. Studies have shown that EPA levels in the organism increase after a restricted diet and that the MAPK signaling pathway mediates the expression of cytochrome P450, helping to drive the metabolic state of the organism [47]. Also, DHA-containing PI can attenuate the induction of *NOS2* to inhibit the activation of p38-MAPK signaling [48]. Since fatty acids significantly alter the metabolism and contractility of vascular smooth muscle and can affect energy metabolism and transduction [49], we suggest that FGFR3 may be negatively regulated in relation to ALA and DHA content. The candidate genes screened in JXY chickens were similarly screened for the mTOR signaling pathway and MAPK signaling pathway.

The above genes were enriched in various pathways related to fatty acid metabolism, and most were enriched in the mTOR signaling pathway. Thus, we hypothesized that the mTOR signaling pathway might be the regulatory pathway affecting FFA content. Also, since *FGFR3* was present among both ALA and EPA candidate genes, and *GRB10* was present among both EPA and DHA candidate genes, in various breeds, we suggest that *FGFR3* and *GRB10* may be the key genes regulating the FFA content in chicken meat.

## Figures and Tables

**Figure 1 foods-12-03988-f001:**
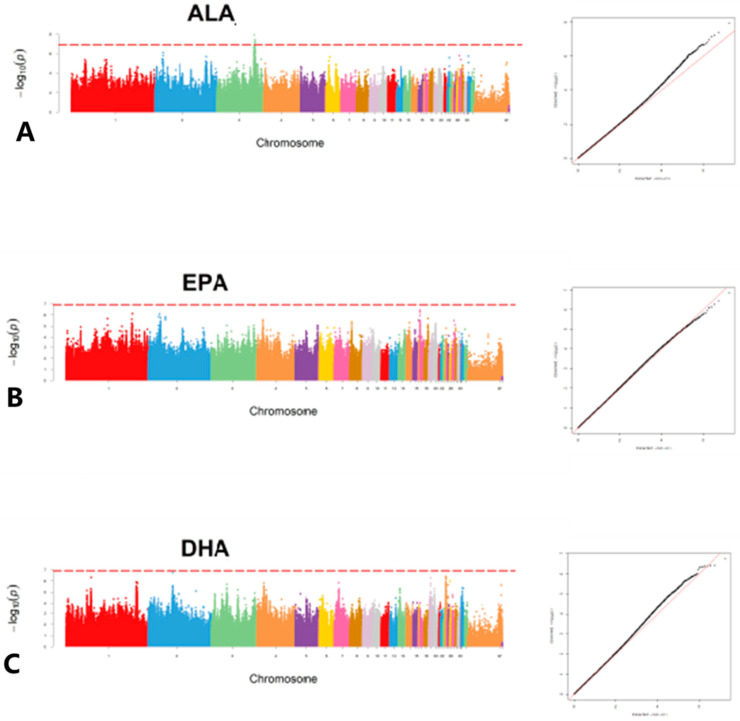
Manhattan and Q-Q plots of FFA GWAS in JXY chickens. (**A**) GWAS analysis of ALA content; (**B**) GWAS analysis of EPA content; (**C**) GWAS analysis of DHA content.

**Figure 2 foods-12-03988-f002:**
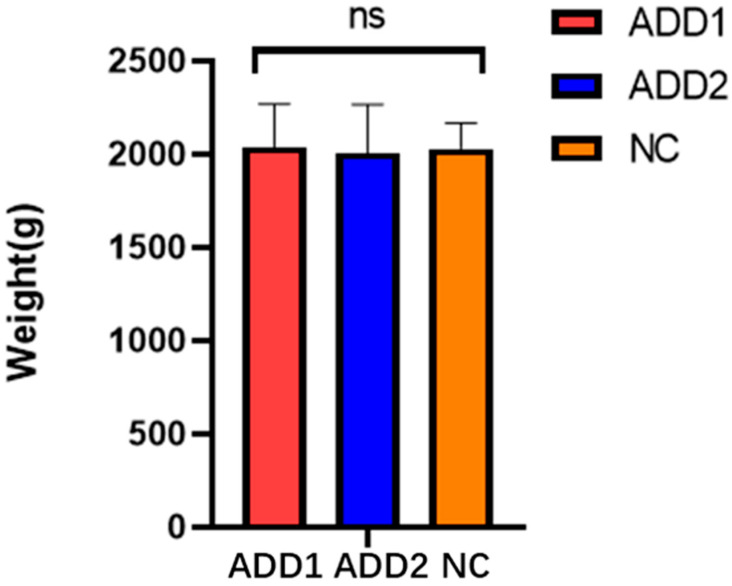
Weight histogram. ADD1 is the feeding group with dietary supplementation of 1% ALA. ADD2 is the feeding group with dietary supplementation of 2% ALA. NC is the feeding group with no dietary supplementation of ALA. ns: shows no significant effect.

**Figure 3 foods-12-03988-f003:**
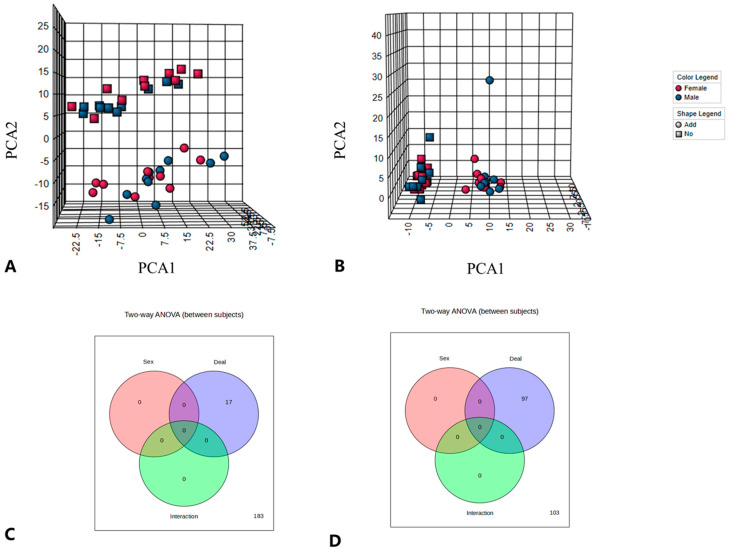
(**A**) Cation PCA score graph; (**B**) anion PCA score; (**C**) cation two-factor test; (**D**) anion two-factor test.

**Figure 4 foods-12-03988-f004:**
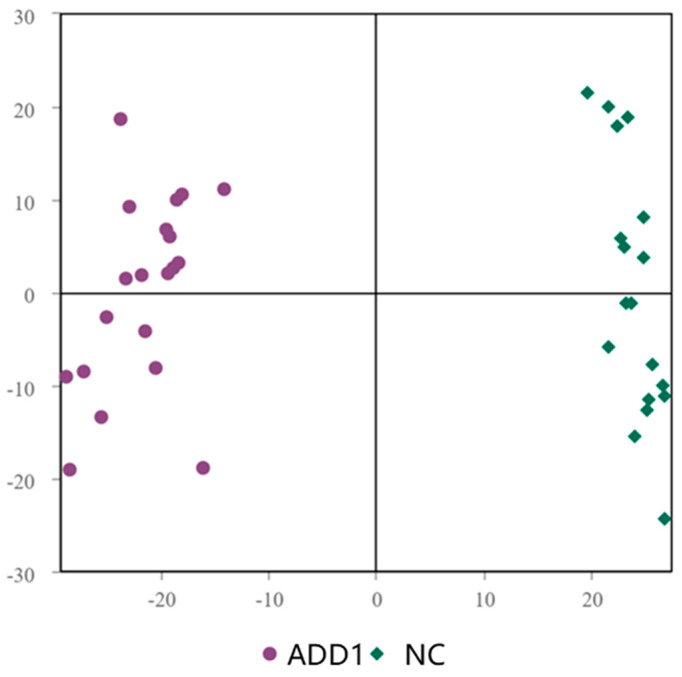
OPL-SDA of ADD1 group with NC group. The ADD1 group and NC group can be distinguished from each other.

**Figure 5 foods-12-03988-f005:**
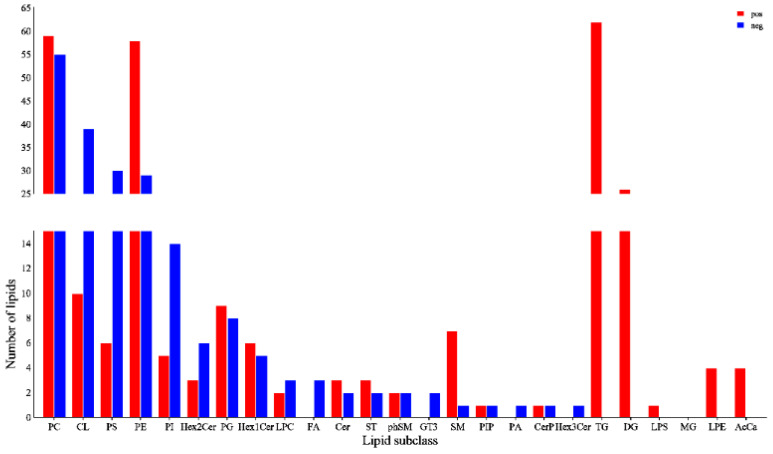
Number of lipid subclasses in different ionic modes. A total of 24 lipid subclasses were identified. There were 206 differential lipid metabolites in the cationic mode and 272 differential lipid metabolites in the anionic mode.

**Figure 6 foods-12-03988-f006:**
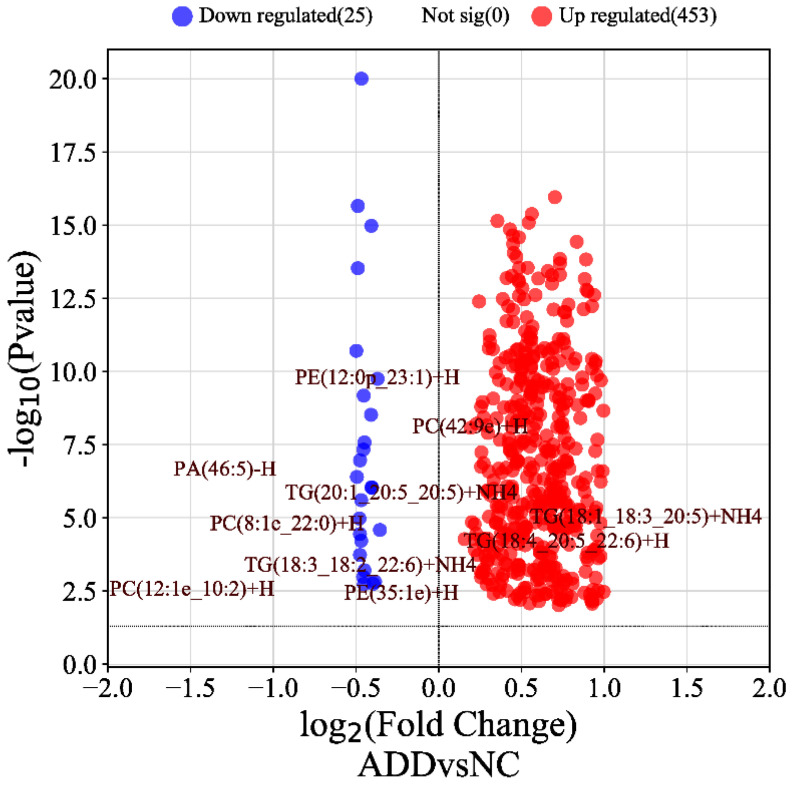
Differential metabolite volcano plot. There were total of 478 differential lipid metabolites in the middle of the ADD1 and NC groups. Of these, 453 upregulated differential lipid metabolites and 25 downregulated differential lipids.

**Figure 7 foods-12-03988-f007:**
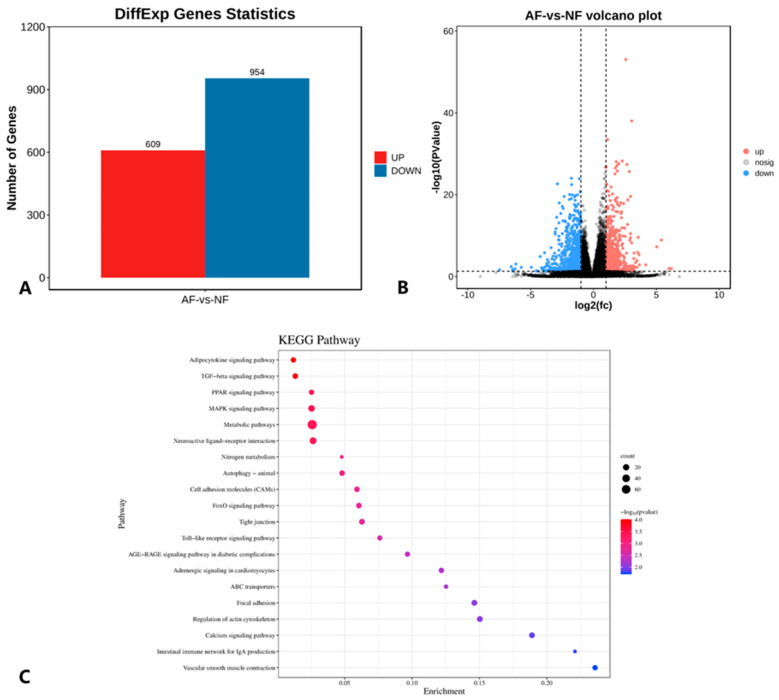
(**A**) Histogram of the number of DEGs between the ADD and NC groups: (**B**) volcano plot of DEGs between ADD and NC groups; (**C**) KEGG enrichment analysis of DEGs between ADD and NC groups.

**Figure 8 foods-12-03988-f008:**
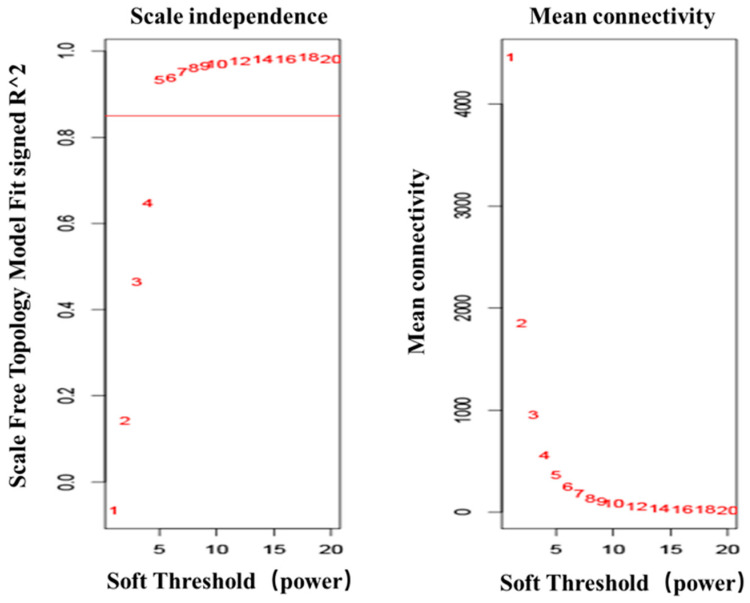
WGCNA optimal soft threshold. Optimal soft threshold = 4.

**Figure 9 foods-12-03988-f009:**
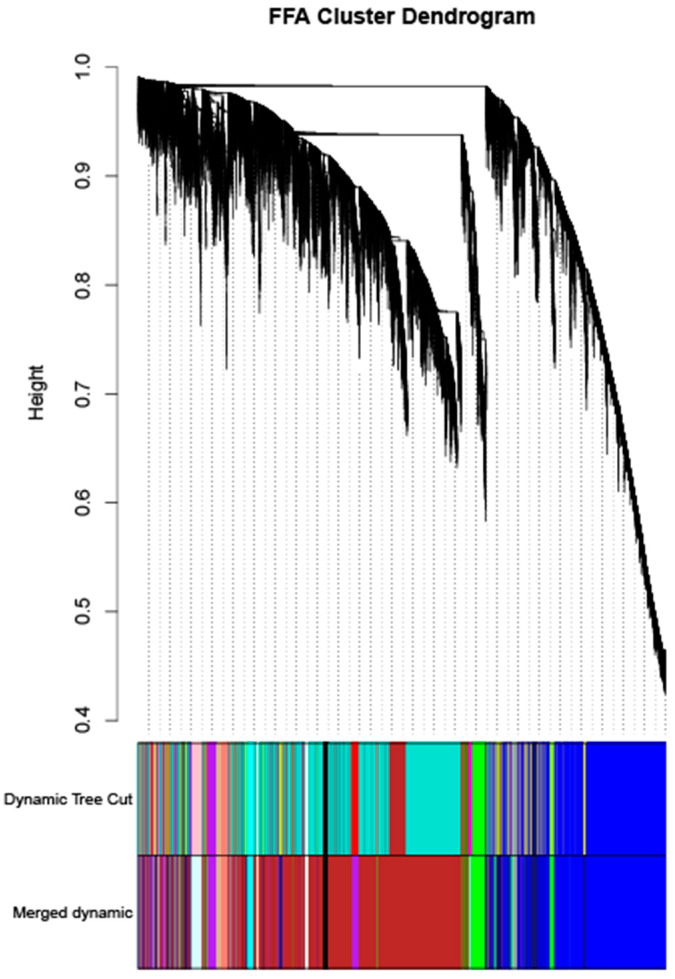
Gene clustering tree (dendrogram).

**Figure 10 foods-12-03988-f010:**
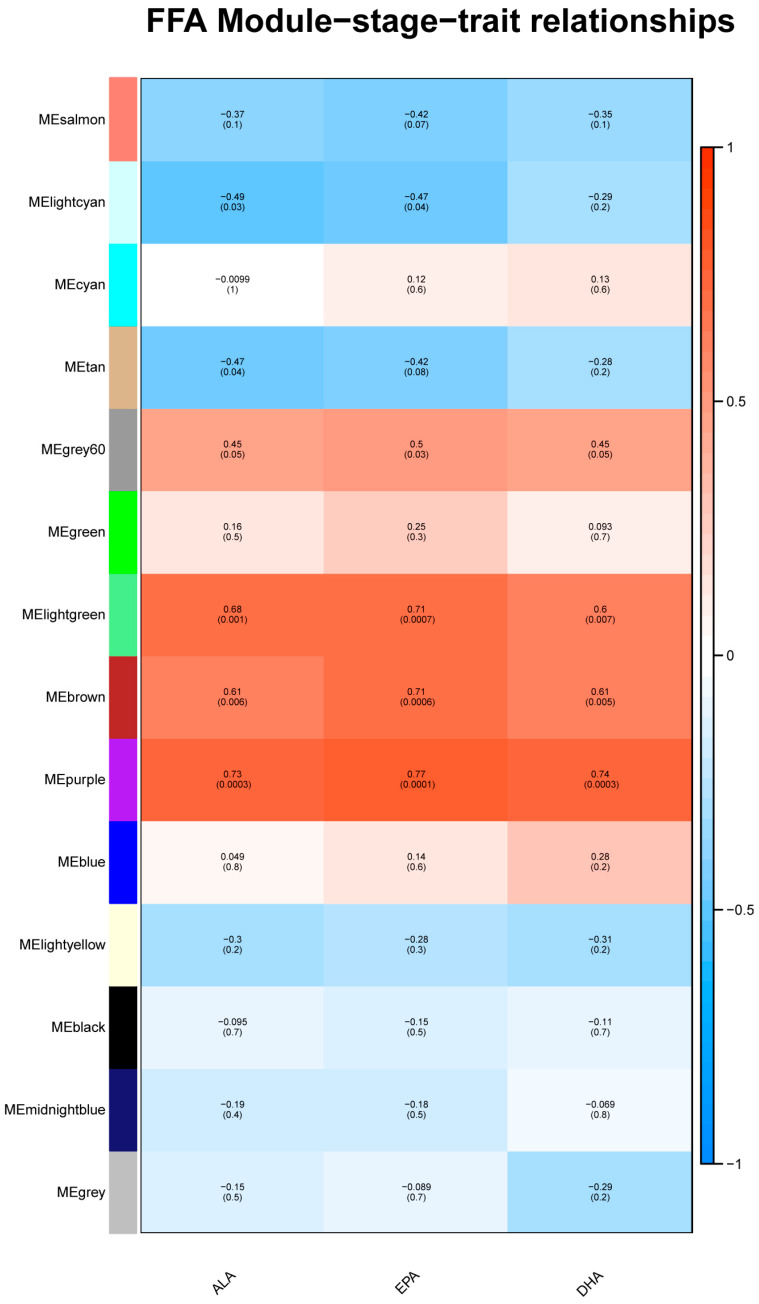
Module related to FFAs. There are 4 significantly relevant modules (grey 60, light green, brown and purple) associated with FFAs.

**Table 1 foods-12-03988-t001:** Table of nutrient composition of diets of different experimental groups.

Nutritional Composition (%)	ADD1	ADD2	NC
Crude protein	≥18.0	≥18.0	≥18.0
Crude fiber	≤9.0	≤9.0	≤9.0
Crude ash	≤9.0	≤9.0	≤9.0
Calcium	0.50–1.30	0.50–1.30	0.50–1.30
Total phosphorus	≥0.40	≥0.40	≥0.40
Calcium chloride	0.20–0.90	0.20–0.90	0.20–0.90
Lysine	≥0.65	≥0.65	≥0.65
Linolenic acid	1	2	0

ADD1 is the composition of feed with 1% ALA supplementation; ADD2 is the composition of feed with 2% ALA supplementation; and NC is the composition of feed without ALA supplementation.

**Table 2 foods-12-03988-t002:** FFA GWAS loci.

Phenotype	Chr	Position	Allele	Allele Frequency	Gene	Coding Area
ALA	3	88,721,989	A/G	0.063	*ENSGALG00000051685*	Non-coding
	3	90,212,152	G/T	0.062	/	/
	3	90,333,836	T/C	0.063	*ENSGALG00000049113*	Non-coding
	3	90,754,923	G/A	0.052	*MYOM2*	Protein coding
	3	90,769,772	T/A	0.061	*MYOM2*	Protein coding

**Table 3 foods-12-03988-t003:** Effect of dietary supplementation with ALA on PL and TG contents.

	Effect of Dietary Supplementation with ALA on PL and TG Contents			SEM	*p* Value
	ADD1	ADD2	NC		
TG (n/mol)	3.85	3.82	4.24	1.86	0.331
PL (n/mol)	4.81	4.33	4.30	0.62	0.000

**Table 4 foods-12-03988-t004:** Effect of dietary supplementation with ALA on fatty acid composition.

Fatty Acid	Fatty Acid Content of Treatment Groups			SEM	*p* Value
	ADD1	ADD2	NC		
C18:2n6c (mg/g)	6.964	7.123	6.920	0.158	0.860
C18:3n3 (mg/g)	1.670 ^b^	2.571 ^a^	0.318 ^c^	0.092	0.000
C20:4n6 (mg/g)	1.588 ^b^	1.418 ^c^	2.287 ^a^	0.042	0.000
C20:5n3 (mg/g)	0.446 ^b^	0.584 ^a^	0.150 ^c^	0.017	0.000
C22:6n3 (mg/g)	0.620 ^b^	0.676 ^a^	0.300 ^c^	0.017	0.000
C18:2n6c (%)	19.654 ^b^	19.327 ^bc^	20.352 ^a^	0.078	0.000
C18:3n3 (%)	4.560 ^b^	6.866 ^a^	0.930 ^c^	0.210	0.000
C20:4n6 (%)	4.720 ^b^	3.960 ^c^	7.009 ^a^	0.133	0.000
C20:5n3 (%)	1.322 ^b^	1.618 ^a^	0.454 ^c^	0.044	0.000
C22:6n3 (%)	1.856 ^ac^	1.873 ^a^	0.919 ^b^	0.046	0.000
ω-6/ω-3	3.24 ^b^	2.42 ^c^	12.21 ^a^	4.93	0.000

Different letters (a–c) indicate significantly different between the treatments (*p* < 0.001).

**Table 5 foods-12-03988-t005:** FFA candidate genes.

Phenotype	Functional Candidate Genes for Fatty Acids	
	Upward	Downward
ALA	*DDX5*, *TBX15*, *MICAL2*, *BEND6*	*FGFR3*, *FAM46C*, *DENND2C*
EPA	*SAMD4A*, *PELI2*, *PIK3CB ZBED1*, *NRAP*, *CA4*, *CDH7*, *ULK1*, *GRK3*, *GRB10*	*FGFR3*, *BEGAIN*, *RFLNA*
DHA	*TRIP11*, *URB2*, *RB1*, *SLC39A10*, *GRB10*, *LONRF3*, *RSPH9*	*CREM*, *PPP1R14C*, *CRISPLD2*

## Data Availability

The data presented in the study are deposited in the Genome Sequence Archive (Wang et al., 2017) in the BIG Data Center (https://bigd.big.ac.cn/gsa/, accessed on 10 July 2021) (Members, 2019), the genomic data accession number CRA002643 and CRA002650.

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
