# Peer review of "Multi-Omics Analysis of Genes Encoding Proteins Involved in Alpha-Linolenic Acid Metabolism in Chicken"

_foods, 2023, doi:10.3390/foods12213988_

Round 1
Reviewer 1 Report
This is an interesting study and well written manuscript on ALA metabolism in chicken. However, before recommending the paper, I would like to make a few points.
Summary – I suggest to the Authors to write a shorter summary, first two sentences are not necessary. According to the instruction for authors, summary should contain maximum 200 words.
Introduction is well written, however it should be checked and some parts must be corrected (double spaces between the words, …). Also, references in the manuscript should be numbered in order of appearance and indicated by a numeral or numerals in square brackets ([1]). This is mandatory to correct in whole manuscript. Moreover, Authors must correct the sentence “A study conducted by Liu et al. in 2022 identified several candidate genes associated with fatty acid deposition in chicken breast meat using a genome-wide association study (GWAS) 6.” and write the reference (Lie et al., 2022) according to Instruction for authors. This also refers on whole manuscript.
Material and method section must be corrected. First of all, I suggest to the Authors to better explain the environmental conditions (“… under the same environmental and nutritional conditions…”) and sample collection. The Tables should be corrected and prepared according to the Instructions. Also, there is no explanation of the Table content, this should be added in footer. I suggest to delete the “group” and wright only “composition”, and also to add the measurement units. Moreover, there are some numbers in table names, this should be checked is it correctly written. Table are not acceptable in current form.
The sentence “Specific methods for the determination of the content and composition of fatty acids are described elsewhere 15.” must be rewritten.
Results section is well written but I suggest to check the English. Also, check the Figure names and occurrence of the numbers within the figure titles.
References in discussion is section must be corrected.
“CRediT authorship contribution statement” must be corrected as “Author Contributions” and rewrite.
Reference section is not written according to the Instruction and must be corrected.
Moderate editing of English language is required.
Author Response
Dear Editor,
Thank you very much for your letter and for the reviewers’ comments concerning our manuscript submitted to Foods (foods-2654142). Those comments are all valuable and very helpful for revising and improving our paper, as well as the important guiding significance to our researches. We revised the paper strictly according to the reviewers’ suggestions. Please find the author’s response in the following context highlighted in red. We hope the revised manuscript is now suitable for publication in Foods
I am looking forward to hearing from you.
Thank you and best regards.
Yours sincerely,
Guiping Zhao
State Key Laboratory of Animal Nutrition, Institute of Animal Sciences, Chinese Academy of Agricultural Sciences
E-mail: zhaoguiping@caas.cn
Reviewer(s)' Comments to Author:
Comments to the Author
This is an interesting study and well written manuscript on ALA metabolism in chicken. However, before recommending the paper, I would like to make a few points.
Summary – I suggest to the Authors to write a shorter summary, first two sentences are not necessary. According to the instruction for authors, summary should contain maximum 200 words.
Response: Thank you for your advice. Changes have been made to the abstract as you suggested. For more information, please see the full article.
Introduction is well written, however it should be checked and some parts must be corrected (double spaces between the words, …). Also, references in the manuscript should be numbered in order of appearance and indicated by a numeral or numerals in square brackets ([1]). This is mandatory to correct in whole manuscript. Moreover, Authors must correct the sentence “A study conducted by Liu et al. in 2022 identified several candidate genes associated with fatty acid deposition in chicken breast meat using a genome-wide association study (GWAS) 6.” and write the reference (Lie et al., 2022) according to Instruction for authors. This also refers on whole manuscript.
Response: Thank you for your advice. Changes have been made to the reference formatting throughout the text. And check and revise the introduction section. and modify the description in the text regarding the findings of Liu et al. For more information, please see the full article.
Material and method section must be corrected. First of all, I suggest to the Authors to better explain the environmental conditions (“… under the same environmental and nutritional conditions…”) and sample collection. The Tables should be corrected and prepared according to the Instructions. Also, there is no explanation of the Table content, this should be added in footer. I suggest to delete the “group” and wright only “composition”, and also to add the measurement units. Moreover, there are some numbers in table names, this should be checked is it correctly written. Table are not acceptable in current form.
Response: Thank you for your suggestions, which have been added to the feeding conditions. Please see the full article for specific information. I modified the formatting of the table to make it look cleaner and clearer.
The sentence “Specific methods for the determination of the content and composition of fatty acids are described elsewhere 15.” must be rewritten.
Response: Thank you for your suggestions. Changes have been made to this sentence, see full text for more information.
Results section is well written but I suggest to check the English. Also, check the Figure names and occurrence of the numbers within the figure titles.
Response: Thank you for your suggestions. The results section has been checked and the figures appearing in the tables and figures have been reconciled. See full text for details.
References in discussion is section must be corrected.
“CRediT authorship contribution statement” must be corrected as “Author Contributions” and rewrite.
Response: Thank you for your suggestions. Changes have been made to the author contributions; please see the full article for more information.
Reference section is not written according to the Instruction and must be corrected.
Response: Thank you for your suggestions. References have been changed to comply with journal requirements. For more information, please see the full article.
Reviewer 2 Report
Multi-omics analysis of genes encoding proteins involved in alpha-linolenic acid metabolism in chicken.
Dear authors,
Thank you for your great efforts, this study investigates the genes encoding proteins involved in alpha-linolenic acid metabolism in chicken meat. The research used genomic data from 477 Jingxing Yellow chickens for a genome-wide association study (GWAS) of fatty acid content traits. Key genes regulating ω-3 fatty acid metabolism in chickens were identified, including FGFR3, GRB10, and signaling pathways (MAPK, mTOR). Upregulation of GRB10 inhibited the mTOR signaling pathway, increasing the content of EPA and DHA. Downregulation of FGFR3 facilitated the conversion of ALA to EPA. The study also analyzed the dose-dependent effects of ALA supplementation on glycerol esters, phospholipid, and fatty acyl contents, as well as the regulatory mechanisms of nutritional responses in FFA metabolism. This provides a basis to identify genes and pathways that regulate the content of FFAs, such as ALA, and offers a reference for nutritional regulation systems in production.
You may find the result of the review in the attached document.
Please check the author details, the information about author names is not complete, please make necessary revision. |
ABSTRACT |
In general, written abstract needs comprehensive revision, in which, the abstract should be written in this following fashion: - Brief introduction and study objectives. - Experimental design and methods, in a more concise and sharp way. - Result trends that also mention the numerical value if possible, or explain the inmprovement or decline following treatments. - Conclusion that includes scientific contribution and industrial application. Please kindly revise the abstract accordingly. |
INTRODUCTION |
||
Line |
Comment |
|
Paragraph 1 |
Many writing errors such as: spacing error, no comma, citation error, etc. still observed in many of introduction sections. Please do thorough proof reading and conduct necessary revision |
|
Paragraph 1 |
Citation is not suiting the given format by the Foods, such as Liu et al. in 2022 that should be followed by citation number, or citation number that is error etc. |
|
Paragraph 3 |
Concept about big food will be more relevant with the inclusion of supporting data about (1) historical background, (2) objectives of policy, (3) its contribution for food supply, etc. the inclusion of the numerical data is suggested in this last paragraph. |
|
Overall, the introduction section has already given necessary point of view and systematic explanation on how this study should be performed. Author need to re-check the format given by the journal and suits the manuscript according to the given guidelines. |
||
MATERIALS AND METHODS |
||
Line |
Comment |
|
2.1. |
Similar comment to that of introduction section, wherein many writing errors such as: spacing error, no comma, citation error, etc. still observed in many of materials and method sections. Please do thorough proof reading and conduct necessary revision. |
|
Table 2.1 |
(?) Please also check for the table format |
|
Materials and methods |
Explanation on why there was different in the number of chickens (477 and 300) |
|
Overall, materials and methods has provided details and supporting references to assure reproducibility. |
||
RESULTS AND DISCUSSION |
Flawless results and discussions are well-shown. Great job. No necessary revision required at this passage. almost all parameters were written in a completely well manner, accompanied by supporting data and references. Authors just need to re-check again on the formatting, i.e. citation format, table format, etc. |
This manuscript requires a thorough proofread to ensure that there is no mistakes on the grammatical and writing.
Also, minor editing on English is required.
Author Response
Dear Editor,
Thank you very much for your letter and for the reviewers’ comments concerning our manuscript submitted to Foods (foods-2654142). Those comments are all valuable and very helpful for revising and improving our paper, as well as the important guiding significance to our researches. We revised the paper strictly according to the reviewers’ suggestions. Please find the author’s response in the following context highlighted in red. We hope the revised manuscript is now suitable for publication in Foods
I am looking forward to hearing from you.
Thank you and best regards.
Yours sincerely,
Guiping Zhao
State Key Laboratory of Animal Nutrition, Institute of Animal Sciences, Chinese Academy of Agricultural Sciences
E-mail: zhaoguiping@caas.cn
Comments to the Author
Dear authors,
Thank you for your great efforts, this study investigates the genes encoding proteins involved in alpha-linolenic acid metabolism in chicken meat. The research used genomic data from 477 Jingxing Yellow chickens for a genome-wide association study (GWAS) of fatty acid content traits. Key genes regulating ω-3 fatty acid metabolism in chickens were identified, including FGFR3, GRB10, and signaling pathways (MAPK, mTOR). Upregulation of GRB10 inhibited the mTOR signaling pathway, increasing the content of EPA and DHA. Downregulation of FGFR3 facilitated the conversion of ALA to EPA. The study also analyzed the dose-dependent effects of ALA supplementation on glycerol esters, phospholipid, and fatty acyl contents, as well as the regulatory mechanisms of nutritional responses in FFA metabolism. This provides a basis to identify genes and pathways that regulate the content of FFAs, such as ALA, and offers a reference for nutritional regulation systems in production.
You may find the result of the review in the attached document.
Please check the author details, the information about author names is not complete, please make necessary revision.
ABSTRACT |
In general, written abstract needs comprehensive revision, in which, the abstract should be written in this following fashion: l Brief introduction and study objectives. l Experimental design and methods, in a more concise and sharp way. l Result trends that also mention the numerical value if possible, or explain the inmprovement or decline following treatments. l Conclusion that includes scientific contribution and industrial application. Please kindly revise the abstract accordingly. |
Response: Thank you for your suggestions. Changes have been made to the abstract; please see the full article for more information.
INTRODUCTION
Line |
Comment |
Paragraph 1 |
Many writing errors such as: spacing error, no comma, citation error, etc. still observed in many of introduction sections. Please do thorough proof reading and conduct necessary revision Response:Thank you for your suggestions. Changes have been made to the introduction; please see the full article for more information. |
Paragraph 1 |
Citation is not suiting the given format by the Foods, such as Liu et al. in 2022 that should be followed by citation number, or citation number that is error etc. Please conduct thorough revision across the manuscript. Response: Thank you for your advice. Changes have been made to the reference formatting throughout the text. For more information, please see the full article. |
Paragraph 3 |
Concept about big food will be more relevant with the inclusion of supporting data about (1) historical background, (2) objectives of policy, (3) its contribution for food supply, etc. the inclusion of the numerical data is suggested in this last paragraph. Response: Thank you for your input. The Big Food View has been added and the rationale inserted. Please see the body of the article for more information. |
Overall, the introduction section has already given necessary point of view and systematic explanation on how this study should be performed.
Author need to re-check the format given by the journal and suits the manuscript according to the given guidelines.
MATERIALS AND METHODS
Line |
Comment |
2.1 |
Similar comment to that of introduction section, wherein many writing errors such as: spacing error, no comma, citation error, etc. still observed in many of materials and method sections. Please do thorough proof reading and conduct necessary revision. Response: Thank you for your advice. Changes have been made to the issues you raised. For more information, please see the full article. |
Table 2.1 |
(?) Please also check for the table format Response: Thank you for your suggestions. The table header has been added. Specific information can be found in the full article. |
Materials and methods |
Explanation on why there was different in the number of chickens (477 and 300) Response: Thank you for your question.The M & M 2.3, 477 individuals were the population used in the preliminary genome-wide association analysis to screen candidate variant sites and annotate genes. In order to verify the effects of the screened loci and genes on ALA, another 300 individuals were selected for ALA supplementation and analyzed by transcriptomics and lipidomics. Finally, the screened genes were further validated by multi-omics joint analysis, which in mentioned in M & M 2.2. |
Overall, materials and methods has provided details and supporting references to assure reproducibility.
RESULTS AND DISCUSSION
Flawless results and discussions are well-shown. Great job.
No necessary revision required at this passage. almost all parameters were written in a completely well manner, accompanied by supporting data and references.
Authors just need to re-check again on the formatting, i.e. citation format, table format, etc.